# Storage Minimization of Marine Energy Grids Using Polyphase Power

Salman Husain *, Gordon G. Parker and Wayne W. Weaver

Department of Mechanical Engineering, Michigan Technological University, Houghton, MI 49931-1295, USA; ggparker@mtu.edu (G.G.P.); wwweaver@mtu.edu (W.W.W.)
* Correspondence: shusain@mtu.edu

**Abstract:** Multiple wave energy converter (WEC) buoys can be used to establish a WEC array-powered microgrid collectively forming a Marine Energy Grid (MEG). An oceanic domain with gravity waves will have significant spatial variability in phase, causing the power produced by a WEC array to have high peak-to-average ratios. Minimizing these power fluctuations reduces the demand for large energy storage by WEC array-powered DC microgrids while also reducing losses in the undersea cable to the shore. Designs that reduce energy storage requirements are desirable to reduce deployment and maintenance costs. This work demonstrates that polyphase power in conjunction with an energy storage system can be used to maintain constant power. This work shows that an *N* WEC array geometry can be designed to reduce the energy storage requirements needed to mitigate the power fluctuations if the WEC array produces constant, polyphase power. Additionally, the conditions that identify the wave frequencies and control the effort needed to produce polyphase power are developed. This paper also shows that increasing the number of WECs in an array reduces aggregate power fluctuations. Finally, WEC array power profiles are investigated using simulation results to verify the mathematical conditions developed for the three and six WEC cases.

**Keywords:** marine energy grid; microgrid; energy storage system; polyphase power; marine structures; wave energy converters (WEC); complementary phase

## 1. Introduction

Innovative marine microgrid solutions can support the energy demands of remote communities, scientific exploration, and establishment of Forward Operating Bases (FOBs). Monitoring and exploration of oceans can be supported by the deployment of offshore remote energy hubs such as WEC arrays and offshore wind farms. Marine Energy Grids (MEGs) can address the aforementioned issues by reducing the dependency on traditional energy grids while expanding the scope of scientific research and exploration. Persistent sensing of offshore environments requires a robust energy supply for powering marine energy hubs serving energy customers such as Ocean Observation Buoys and Underwater Unmanned Vehicles (UUVs).

A microgrid is a localized and controllable power grid that maintains a bus voltage, using energy generation assets and energy storage systems (ESS) while serving electrical loads demanded by customers [1,2]. An islanded microgrid is isolated from any other grid as it operates at its most cellular level [1,2]. Islanded microgrids can be networked with other microgrids or to the general grid. Such a network can be a modular network with interchangeable and switchable modules depending on the energy demands [3].

A significant hurdle in mainstream adoption of marine energy is the lack of consensus and research in grid integration strategies [4,5]. While wind and solar energy devices have largely achieved design convergence facilitating their grid integration strategies, marine energy and, especially, wave energy research is still exploring different devices and their respective grid integration [6]. Preziuso et al. presented a literature review on grid

integration of marine energy assets [5]. They surveyed the grid integration viability for wave energy, tidal energy, and ocean current energy assets. They further identified that the grid integration of marine energy assets requires arrays of multiple marine energy assets integrated as microgrids. The fluctuation of waves in nature causes power and voltage fluctuations in a WEC's Power Take-Off (PTO). The power fluctuations from a WEC can be mitigated by WEC array control [4], aggregation of a large number of WECs in an array [5], and using energy storage systems (ESS) [7]. WEC arrays that produce constant power are desirable as they (1) reduce storage requirements and (2) reduce transmission losses.

Sjolte et al. showed that ESS could reduce the WEC array's power fluctuation by as much as 18% [8]. They recognized that ESS is essential in mitigating power fluctuation in a MEG. They also analyzed the Levelized Cost of Energy (LCOE) and showed a cost analysis of using a battery as the ESS for a MEG operating at six different sea-states. Yu et al. showed that power smoothing techniques could reduce power fluctuation in a MEG by an order of magnitude and therefore can have a direct effect on the LCOE [9].

The mitigation of power fluctuation using ESS for the WEC array microgrid can be accomplished using a battery as investigated by Stefek et al. [10], a super-capacitor as proposed by Brando et al. [7], or a hybrid of battery and super-capacitor as modeled by Parwal et al. [11]. Brando et al. base their ESS selection on power quality, bridging power, and energy management [7]. They also considered the charging and discharging response times of various ESS so that the microgrid's bus voltage can be maintained during power shortfalls. They control the microgrid such that the ESS is connected to the DC side of their circuit through a buck-boost chopper as the DC side interfaces with a three-phase AC grid (AC side) using a Voltage Sensitive Relay (VSR). The VSR control ensures that the active power transfer with the grid from the DC side is as constant as possible while the ESS absorbs the excess power produced instantaneously by the WECs or compensates for the shortfalls as needed to maintain the bus voltage. Zhou et al. presented a numerical framework for sizing ESS for WEC applications [12]. They also show that the dynamics and the motion-control of the WEC and control strategy of the electrical drive must be integrated to accurately model the ESS [12].

The physical nature of oceanic waves entails phase offsets in the wave elevations received at individual members of a WEC array as a function of array layout. Rollano et al. quantified the effect of phase information on the power output of a virtual WEC array [13]. They compared the power output performance of a WEC array in a phase-averaging wave model using Simulating WAves Nearshore (SWAN) against a phase-resolving wave model FUNWAVE-TVD [14,15]. A phase-averaging wave environment can be used for a WEC array with a large number of WECs because each WEC at a different location in the array has the opportunity to sample a different phase of the incoming wave to reduce the variability in the aggregate power [13]. However, they point out that power systems are vulnerable to large wave amplitudes, making a phase-resolved wave environment crucial in avoiding underestimating such wave events. Rollano et al. concluded that the phase-resolving wave environment is especially apposite for modeling arrays with a small number of WECs.

Tidal energy researchers have identified the theoretical advantages of using multiple tidal energy assets installed such that they produce staggered power profiles at mutually complementary phase differences [5]. The advantage thereof is that the sum of the power profiles generated at staggered phases is a relatively smooth power profile that can theoretically be a constant flat-line power profile. Giorgi and Ringwood used a multi-objective optimization to evaluate eleven tidal energy sites around Ireland such that the optimization simultaneously maximized the mean power and the minimum power while minimizing variance by minimizing the variance in the total power using two variables at each site: number of installed turbines and type of installed turbines [16]. Preziuso et al. remarked that the multi-objective optimization used by Giorgi and Ringwood often have conflicting objectives. Clarke et al. also acknowledged the practical challenges in generating staggered power profiles using tidal energy assets [17].

Interestingly, the phase profile of gravity waves varies over much more local scales when compared with the phase profiles of tidal energy harvesting sites. This implies that the staggered phase profiles received at each WEC in an array can be capitalized such that the phase of the power produced by each member of the WEC array is complementary to each other, and the array produces constant power.

Renewable energy assets produce stochastic power, and therefore, many researchers have used DC microgrids for their grid integration [2,3,18]. Cook et al. showed the advantages of using a mode-adaptive control scheme over decentralized N-state droop control, either of which can be a compelling control strategy as a localized WEC array control law [2]. A bi-directional flow of power between the WEC array and the microgrid can further enhance the power produced by the WEC array using control action. Forehand et al. observed that not much work has been carried out on the coupling of WEC arrays and electrical grids and even less work has been conducted on a bi-directional coupling between a WEC array and electrical grid [4]. They introduced a bi-directional WEC array model that was coupled with a microgrid. Their bi-directional model can be used to identify individual and collective contributions to the power quality as well as the effect of the power faults in the microgrid on the PTO. Active control of WEC arrays supplemented with ESS can further mitigate voltage fluctuation in the microgrid [5]. In [19], Weaver et al. acknowledges the need for a large energy storage capacity for DC microgrids with renewable power sources to mitigate power fluctuations. In [18], Weaver et al. integrated ESS in their DC microgrid and illustrated the advantages of a novel approach to minimize local storage capacity by using droop control actuated by the local ESS. Wilson et al. in [20] and Weaver et al. in [21,22] showed WEC array networked DC microgrids that used HSSFC. HSSFC uses the current from the ESS to implement a feedback controller in conjunction with feedforward control using the reference current on the bus side of the microgrid. They show that producing polyphase power minimizes the power fluctuations, which in turn minimizes the size of the ESS.

More recently, Latif et al. presented a comparison of various power management control strategies for MEGs [23]. They compared the performance of PID-based controllers with the performance of optimization-based algorithms such as the genetic algorithmic technique (GA), particle swarm technique (PSO), firefly algorithmic technique (FA), cultural algorithmic technique (CA), and the recent meta-heuristic grasshopper algorithmic technique (GOA). Their power management optimization aims for robust load frequency control. Tarasiuk et al. identified the advantages of using a Hilbert–Huang transform (HHT) controller to mitigate the frequency of variation in MEG loads and power generation [24]. Optimization-based approaches for better power management were extended to a multi-objective optimization by Zhou et al. [12]. They investigated hybrid MEGs that have a mixture of marine energy assets such that the MEG contains wind energy and tidal energy assets and demonstrated that multi-objective optimization strategies can respond to load and generation variability more efficiently than single-objective optimizations. Fayek et al. also investigated a hybrid MEG comprising wave energy and wind energy assets [25]. They showed that their load frequency control algorithm can counter the undesirable effects of variability in both loads and power generation characteristics. They discussed the advantages of deloading, where the MEG supplies less power than it actually can to replenish energy storage buffers.

As discussed above, a wide variety of MEG power control strategies try to counter the effects of this variability using either single-objective or multi-objective optimization schemes while reducing the energy storage capacity required to maintain a constant power supply. .

## 2. Article Contributions

The motivation of this paper is to explore ways to achieve constant power through design and WEC force control, thereby facilitating the grid integration of WEC arrays. The ESS can be supplemented by control action in such a way that the power injections

from the WEC array are maximized while at the same time ensuring that the power fluctuations are minimized.

This paper hypothesizes that the power fluctuations in the aggregate power produced by a WEC array can be minimized by placing the individual WECs in the array at locations that receive the incoming wave at staggered phase differences. If the phase offsets amongst individual WECs results in polyphase power, the phases of the powers they produce individually are staggered, but the aggregate sum of these powers is constant.

This paper develops strategies that can help reduce the energy storage requirements of MEGs. This paper recommends that the polyphase condition should be a design parameter for WEC array layout and control. The recommendations made in this paper can be easily integrated with PTO force control strategies. This paper proposes that the WEC array PTO force control should be informed by the advantages of producing polyphase power. This paper demonstrates that a WEC array that produces polyphase power will require a smaller ESS and will require less controller effort when compared with nonpolyphase power.

The rest of this paper is organized as follows; Section 3 develops the mathematical conditions needed to design a WEC array that produces polyphase power. A simulation case study is presented in Section 4. Finally, conclusions are made in Section 5.

### 3. Constant Power WEC Array Conditions

This section develops the spacing, control force phasing, regular wave frequency, and energy storage control conditions so that an ideal WEC array can produce constant power. The most general case is considered first, followed by a homogenous array in which sufficient conditions for constant power are greatly simplified.

Consider the *N* WEC array of Figure 1, where the devices are arranged such that their hydrodynamic coupling is negligible and they all receive the same wave forces. The vertical speed of the *i*th WEC is assumed to be

$$\dot{z}_i = v_i \cos(kx_i - \omega t) \tag{1}$$

where $v_i$ is its amplitude, $k$ is the wave number, $x_i$ is the $x$ component of its position vector from the origin of the reference frame, $\omega$ is the temporal frequency, and $t$ is time. The wave number and frequency are related by the dispersion equation of Equation (2)

$$\omega^2 = gk \tanh(kh) \tag{2}$$

where $g$ is the local gravitational acceleration and $h$ is the water depth. Assume that the *i*th WEC's control force is

$$F_{c,i} = k_{c,i} v_i \cos(kx_i - \omega t + \theta_i) \tag{3}$$

where $k_{c,i}$ scales and $\theta_i$ phase shifts the velocity. The *i*th WEC power output is

$$p_i = k_{c,i} v_i^2 \cos(kx_i - \omega t) \cos(kx_i - \omega t + \theta_i), \quad 0 \leq \theta_i \leq \frac{\pi}{2} \tag{4}$$

where $\theta_i = 0$ yields maximum power and $\theta_i = \frac{\pi}{2}$ results in minimum power. The control law of Equation (3) captures a wide range of control strategies, including the resistive approach used in the case studies in Sections 3 and 4. Summing Equation (4) over all $N$ WECs, employing some trigonometry and defining $2A_i = k_{c,i} v_i^2$, the array power is

$$p = \sum_{i=0}^{N-1} A_i [\cos(2(kx_i - \omega t) + \theta_i) + \cos \theta_i] \tag{5}$$

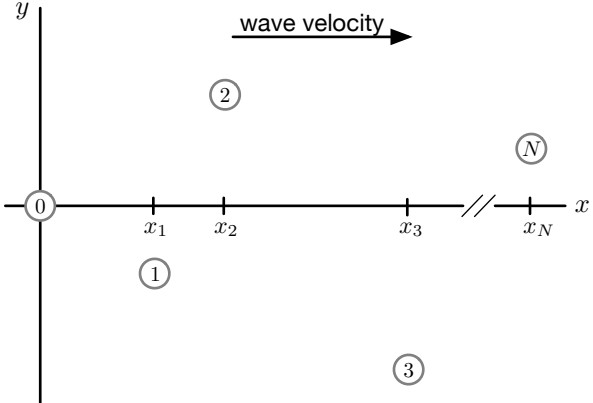

**Figure 1.** Top down view of an $N$ buoy WEC array. WEC motion is assumed to be in heave only.

To achieve constant power, $\dot{p}$ must be zero for all times leading to the condition

$$\sum_{i=0}^{N-1} A_i \sin\left(2(\phi_i - \omega t)\right) = \cos 2\omega t \sum_{i=0}^{N-1} A_i \sin 2\phi_i - \sin 2\omega t \sum_{i=0}^{N-1} A_i \cos 2\phi_i = 0 \qquad (6)$$

where

$$\phi_i = kx_i + \frac{1}{2}\theta_i \qquad (7)$$

This results in two simultaneous equations, Equation (8), that, when satisfied, ensures the array power is constant. For a given regular wave, $k$ or $\omega$, and specified WEC spacing, $x_i$, there are $2N$ free parameters—$A_i$ and the $\theta_i$ of Equation (7).

$$\begin{aligned}
\sum_{i=0}^{N-1} A_i \sin 2\phi_i &= 0 \\
\sum_{i=0}^{N-1} A_i \cos 2\phi_i &= 0
\end{aligned} \qquad (8)$$

If the WEC spacing is fixed, then these constant power conditions are satisfied for only a set of specific wave numbers $k$. This may seem limiting; however, Equation (7) illustrates that, by adjusting the WEC control phase, $\theta_i$, it is possible to achieve constant power for a continuous variation in $k$ with some sacrifice of output power. Reducing an individual WECs power output by adjusting its $\theta_i$ may be justifiable since maintaining the array's constant power reduces (1) power loss due to storage cycling, (2) storage capacity requirements, and (3) losses arising from transmitting sinusoidal power from the array.

To quantify the array storage requirements, consider the case where Equation (8) is not satisfied. The array's power output due to the WECs, Equation (5), will be sinusoidal. A storage device can be added to the array that absorbs and contributes energy such that the net power transmitted from the array is again constant. Considering the storage device as the $N$th "WEC" in the array, the polyphase conditions of Equation (8) become

$$\begin{aligned}
\sum_{i=0}^{N-1} A_i \sin 2\phi_i &= -A_N \sin 2\phi_N \\
\sum_{i=0}^{N-1} A_i \cos 2\phi_i &= -A_N \cos 2\phi_N
\end{aligned} \qquad (9)$$

The magnitude and phase of the storage power, $A_N$ and $\phi_N$, are

$$(2A_N)^2 = \left( \sum_{i=0}^{N-1} A_i \sin 2\phi_i \right)^2 + \left( \sum_{i=0}^{N-1} A_i \cos 2\phi_i \right)^2$$

$$\tan \phi_N = \frac{\sum_{i=0}^{N-1} A_i \sin 2\phi_i}{\sum_{i=0}^{N-1} A_i \cos 2\phi_i} \tag{10}$$

where $A_N$ can be used to determine the storage capacity based on the expected wave frequency, array configuration, and the use of $\theta_i$ to trade-off power output for storage capacity.

As an example, consider an $N = 5$ array where the WECs have identical power output, $A_i = A$, when operating at peak capacity, $\theta_i = 0$. For a wave frequency $\omega = \omega_0 = 0.3237 \, \text{rad/s}$, or $k = 0.0126 \, \text{m}^{-1}$ where the depth is $h = 100 \, \text{m}$, the total power is constant when $\phi_i = \{0°, 72°, 144°, 216°, 288°\}$ for $i = 0 \ldots 4$, as shown in Figure 2a.

Now, consider the case where the wave frequency is reduced by 20%, $\omega = 0.8\omega_0$, but the WEC spacing is unchanged, resulting in the sinusoidal total power output shown in Figure 2b. In this example, $x_i = 100i \, \text{m}$, but any appropriate spacing would suffice. The array can be brought back to constant power in two ways: (1) through storage control or (2) by adjusting the control force phasing, $\theta_i$, of Equation (3). Figure 2c shows the storage solution using Equation (10) where $A_N = 0.6215$. The required storage capacity, $E$, is the magnitude of the integral of the storage power, in this case $E = A_N/0.8\omega_0$.

Another approach to securing constant power is to control the phase angle $\theta_i$. These can be found by solving a constrained optimization problem: calculate the $\theta_i^*$ that minimizes $J = \sum \theta_i$ subject to the equality constraints of Equation (8) and $0 \leq \theta_i \leq \frac{\pi}{2}$. The motivation for this particular $J$ is to keep the individual WEC output as close as possible to the maximum. It is important to note that alternative cost functions can be used that may be better from an array perspective. The results for $J = \sum \theta_i$ are shown in Figure 2d where $\theta_0^* = \theta_3^* = 0$ and $\theta_1^* = 52.5°$, $\theta_2^* = 11.4°$, and $\theta_4^* = 30.0°$. While constant power was achieved, the array power was reduced from 2.5 to 2.3 or about 8%. This could be an acceptable reduction depending on the availability of array storage or the loss that would be incurred transmitting the sinusoidal power of Figure 2b. It is important to note that, as the number of WECs in the array increases, the array power penalty decreases dramatically. For example, if 10 WECs are used, then the power reduction is about 0.1% for the same 20% change in wave frequency.

Next, consider the special case where all the WEC's are identical, $A_i = A$. Without any loss of generality, the reference frame origin can be placed at the 0th WEC, as shown in Figure 1. The constant power conditions in this situation are

$$\sum_{i=1}^{N-1} \sin 2\phi_i = 0$$

$$\sum_{i=1}^{N-1} \cos 2\phi_i = -1 \tag{11}$$

For $N = 3$, the solution is $\phi_1 = 120°$ and $\phi_2 = 240°$, and for $N > 3$, there are multiple solutions. Using Equation (7), the $\phi_i$ can be attained using a variety of spacing, $x_i$, and control phasing, $\theta_i$.

Applying two additional conditions, (1) equal spacing, $x_i = Li$, where $L$ is the distance between any two WECs, and (2) peak power operation, $\theta_i = 0$, the constant power sufficient conditions can be written using closed form set operations. To show this, arithmetic progression expressions are substituted into Equation (11), as shown in Equation (12).

$$\sum_{i=0}^{N-1} \sin(2kLi) = \frac{\sin(NkL)}{\sin(kL)} \sin((N-1)kL) = 0$$

$$\sum_{i=0}^{N-1} \cos(2kLi) = \frac{\sin(NkL)}{\sin(kL)} \cos((N-1)kL) = 0$$

(12)

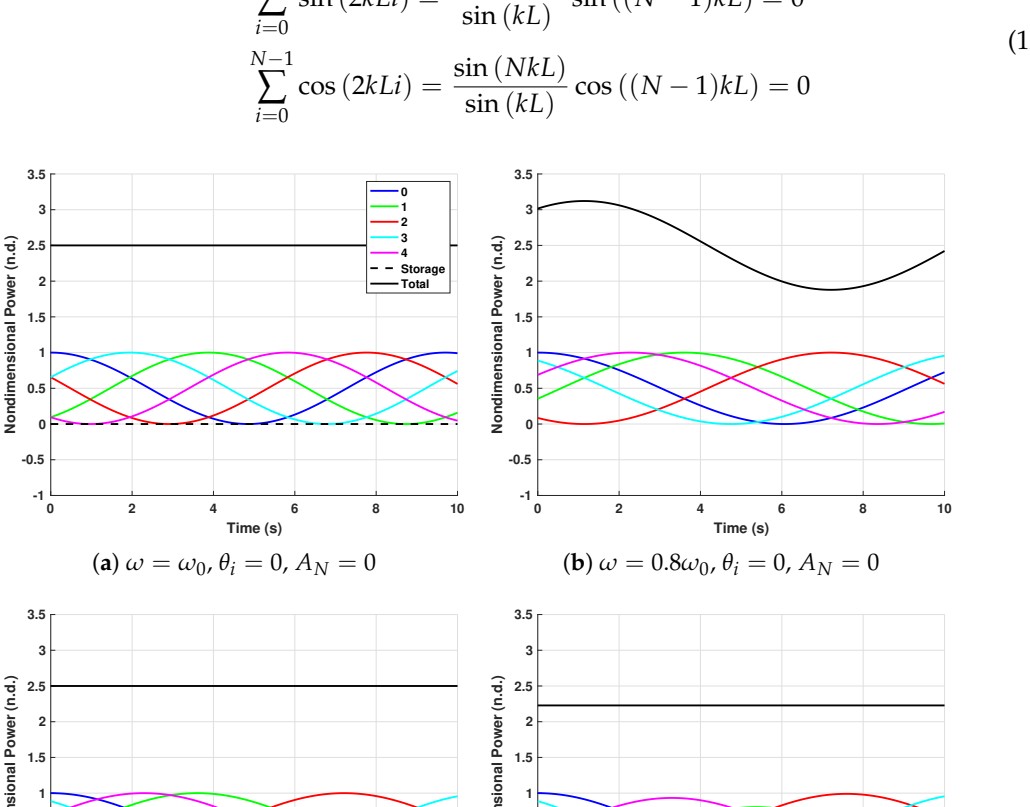

(**a**) $\omega = \omega_0$, $\theta_i = 0$, $A_N = 0$　　　　　　(**b**) $\omega = 0.8\omega_0$, $\theta_i = 0$, $A_N = 0$

(**c**) $\omega = 0.8\omega_0$, $\theta_i = 0$, $A_N = 0.6215$　　　(**d**) $\omega = 0.8\omega_0$, $\theta_i = \theta_i^*$, $A_N = 0$

**Figure 2.** Five-WEC array examples illustrating (**a**) constant power when the wave frequency, $\omega_0$, satisfies the polyphase conditions; (**b**) the effect on power when $\omega \neq \omega_0$; (**c**) the use of storage to achieve constant power; and (**d**) the use of $\theta_i$ to achieve constant power without storage.

If the wave length, $\lambda = \frac{2\pi}{k}$ is known, Equation (12) can be manipulated to create the sufficient conditions for the set of separation distances, $\{L\}$,

$$\{L\} = \left\{\frac{m\lambda}{2N}\right\} - \left(\left\{\frac{m\lambda}{2(N-1)}\right\} \cup \left\{\frac{m\lambda}{2}\right\}\right) \quad m = 1\ldots M$$

(13)

where the subtraction symbol is the relative compliment operator; $M$, $R$, and $Q$ are $\geq 1$; and wavelength and wavenumber are related by $\lambda = \frac{2\pi}{k}$. The number of separation distances returned is $M$. Similarly, if the separation distance $L$ is specified, then the sufficient condition for the set of wavelengths, $\{\lambda\}$, that yield constant power are

$$\{\lambda\} = \left\{\frac{2NL}{m}\right\} - \left(\left\{\frac{2(N-1)L}{m}\right\} \cup \left\{\frac{2L}{m}\right\}\right) \quad m = 1\ldots M$$

(14)

Figure 3 helps to visualize the possible spacing solutions given a wave frequency $\omega_0$ for the five-WEC array considered earlier. In this case, the wavelength was set to $\lambda_0 = 60\,\text{m}$ corresponding to $\omega_0 = 1.1036\,\text{rad/s}$ when the water depth is $h = 100\,\text{m}$. The most densely packed solution is shown with blue circles, and the most sparse solution is shown with red circles. They both have the $0th$ WEC in common at the origin. The horizontal lines are the phase angles calculated earlier and the diagonal lines repeat according to the wavelength.

Valid solutions must span all five phases, $\phi_i = \{0°, 72°, 144°, 216°, 288°\}$ for $i = 0\ldots4$ while intersecting the diagonal wavelength lines.

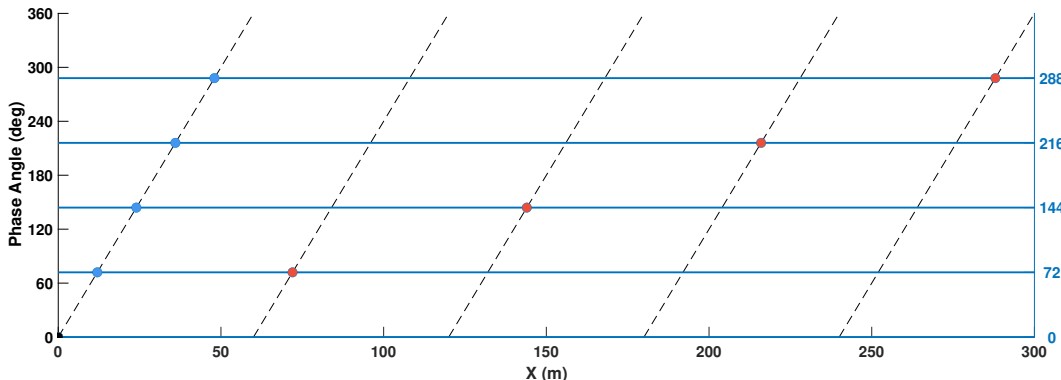

**Figure 3.** Two spacing solutions for the five-WEC array example introduced earlier. The phase, $\phi_i$ described by Equation (8), is plotted with respect to the buoy location $x_i$. The 0th WEC for both solutions at the origin is shown as a black circle. A tightly packed solution is shown with blue circles, while a sparsely packed solution is shown in red.

### 4. Simulation Case Study

While the previous development was motivated by the WEC array application, it did not consider the WEC's dynamic response. Thus, it was applicable to any set of $N$ generators but ignored the more realistic WEC response to waves. In this section, constant power conditions are explored using WEC array dynamic models. Three and six-WEC arrays were considered for the special case described above, where their inter-WEC spacing, $L = 100$ m, was constant and their power outputs were identical, $A_i = A$. The WEC array simulation included a storage controller that ensured the total power was constant regardless of the wave frequency or spacing. The simulation was executed by sweeping through a range of wave frequencies where the storage power sinusoidal amplitude was used to assess performance. Similar to the analysis above, the six-WEC array was less sensitive to variations in the conditions that yielded constant power than the three-WEC array.

The WEC array model consisted of $N$ cylindrical buoy point absorbers, each with 1.0 m radius and 1.0 m draught. It was assumed that their constant, inter-WEC spacing $L$ was sufficiently large so there was no dynamic coupling. Furthermore, all of the WECs incident waves had the same properties and so their power outputs were identical as mentioned above. The local water depth was $h = 100$ m.

The $i$th WEC's dynamic model was adapted from [26,27] and is shown in Equation (15), where the $i$ subscripts are omitted for brevity.

$$(m + a_\infty)\ddot{z} = F_e + F_c + F_r + F_s \tag{15}$$

The added mass at the infinite frequency is denoted by $a_\infty$, and the forces on the right are excitation, $F_e$, control, $F_c$, radiation, $F_r$, and hydrostatic $F_s$. The expressions for each are given in Equation 16.

$$
\begin{aligned}
F_e &= \int_{-\infty}^{\infty} \Big[ h_{exc}(\tau)\eta(t-\tau) \Big] d\tau \\
F_c &= -k_c \dot{z} \\
F_r &= -\int_{0}^{t} h_r(t-\tau)\dot{z}\, d\tau \\
F_s &= -k_s z
\end{aligned}
\tag{16}
$$

where $h_{exc}$ is the excitation impulse response function, $\eta(t - \tau)$ is the wave elevation, $h_r$ is the radiation impulse response function, and $k_s$ is the linear hydrostatic stiffness constant. $z$ and $\dot{z}$ correspond to the vertical WEC displacement and velocity, respectively. The hydrostatic stiffness constant and other hydrodynamic coefficients were generated using the boundary element solver in the hydrodynamic analysis software WAMIT. $h_{exc}$ and $h_r$ were computed using the Fourier-transform of the frequency-dependent excitation force and radiation damping hydrodynamic coefficients, respectively, and are shown in Figure 4. Similarly, the added mass at infinite frequency was found to be $a_\infty = 1950.6$ kg, and the hydrostatic stiffness was found to be $k_s = 3.1$ kN/m. The rate feedback control law, $F_c$, is a subset of the family introduced earlier in Equation (3). The excitation force contains the dynamic Froude–Krylov and diffraction forces. The linear assumptions require that the incoming waves have small amplitude and steepness and that the WEC motions are also small.

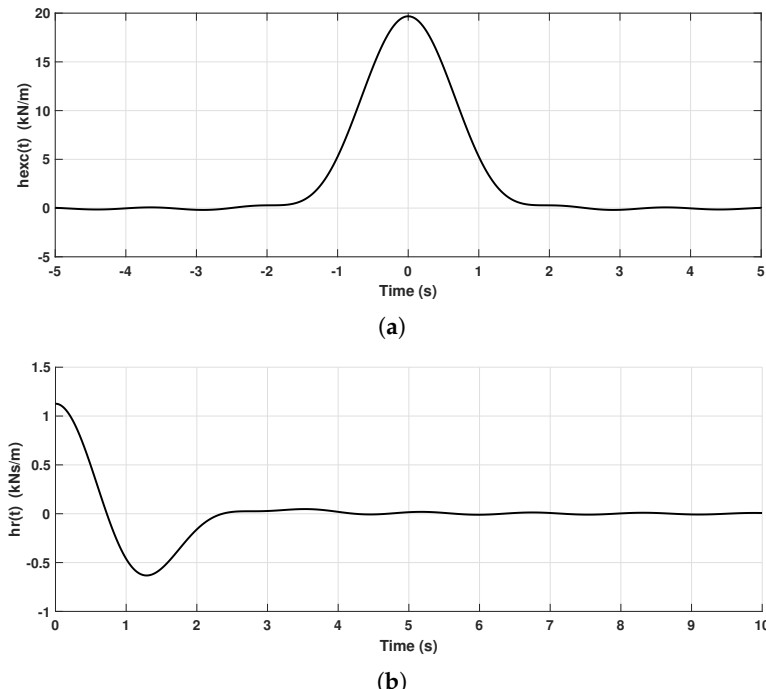

**Figure 4.** The impulse response functions experienced by each WEC. The inter-WEC spacing was $L = 100$ m, and no significant hydrodynamic coupling could be observed for the cylindrical WECs with 1.0 m radius and 1.0 m draught. (**a**) The excitation force impulse response, $h_{exc}$. function; (**b**) the radiation force impulse response, $h_r$, function.

The peak RMS power and energy were calculated for the last 100 s of the simulations (total run time being 500 s) to avoid the initial transient behavior. The peak RMS power as shown in Figure 5 was calculated using the storage power sinusoidal amplitude in Equation (10). The energy spent by the ESS was calculated by integrating the storage power over the last 100 s, which in this case would be $E = A_N/\omega$. Notice, in Figure 5, that the ESS power increases with an increase in wave frequency while the ESS energy is less sensitive to increases in wave frequency because, as the power increases, so does the wave frequency. Therefore, the ESS specifications should be informed by both the power and energy requirements.

The $i$th WEC's power output is given by Equation (4), with $\theta_i = 0$, and the total WEC power by Equation (5). When the array conditions are such that constant power is achieved, the total power is $p = \frac{1}{2}NA$. This value was used as the reference for the storage controller, which simply added the necessary power to make up the difference between the reference and the instantaneous, total WEC power output.

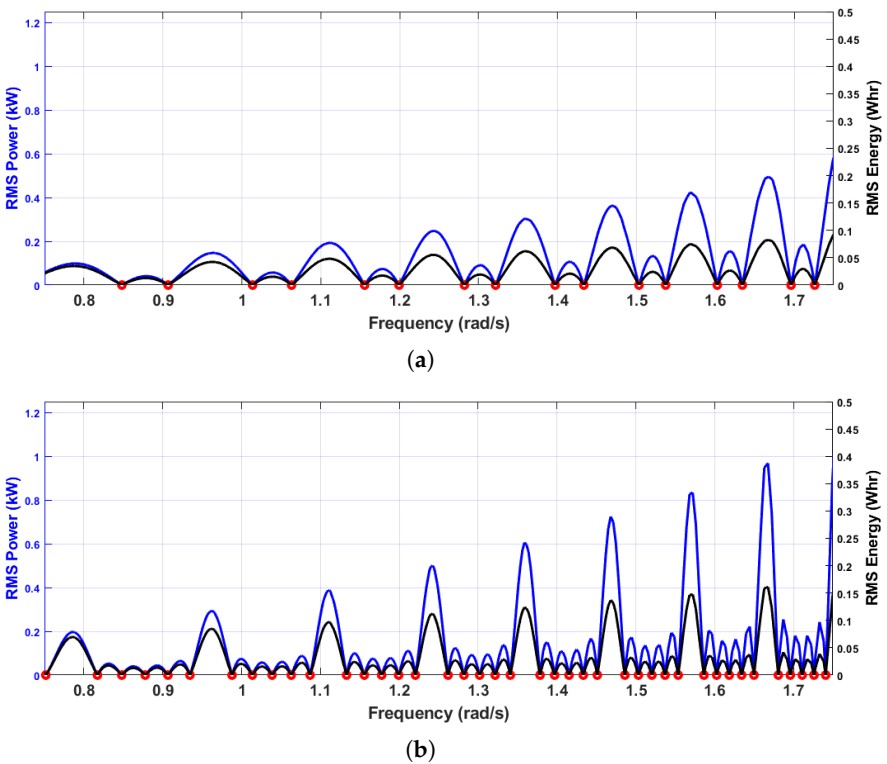

**Figure 5.** Storage power and energy, as a function of wave frequency (0.75 rad/s–1.75 rad/s), required to ensure that the WEC array power was constant for both three and six-WEC arrays. The inter-WEC spacing was $L = 100$ m. The storage energy requirements are less sensitive to increases in wave frequency when compared with storage power. (**a**) Three-WEC array and (**b**) six-WEC array.

For both the three-WEC and six-WEC arrays, the simulation was used to assess the storage needed to produce constant power with inter-WEC spacing held at $L = 100$ m. The incident wave amplitude was 0.25 m, and the wave frequency was incremental for a total of 1000 frequencies between 0.75 to 1.75 rad/s . At each frequency, the simulation was allowed to run for 500 s. The simulated storage power for both cases are shown in Figure 5. The constant power frequencies are shown as red dots as computed by the $\{\lambda\}$ set of Equation (14). As expected, these match well with the minima of the simulated storage power.

One of the primary differences between the three- and six-WEC arrays is their sensitivity to variations in wave frequency. If the objective is to reduce the amount of oscillation in the array power, the least desirable situation is when the wave frequency corresponds to a local maximum in the storage plots. Selecting a spacing that avoids the maxima for the expected wave frequency variation is more easily achieved for the six-WEC array. This storage "flatness" increases dramatically with $N$.

## 5. Conclusions

Energy generated by a single wave energy converter needs modulation, but a network of WECs with staggered phases can complement each other to result in constant power that can be integrated to a marine energy grid. Increasing the number of WECs in a WEC array reduces the fluctuations in the aggregate power produced by the WEC array. If the WECs constituting the WEC array produce polyphase power, the fluctuations in the net power produced by the WEC array can be mitigated. This work shows that the phase of the ocean waves can inform the layout of a WEC array to produce polyphase power. The simulation model also demonstrates that the control effort to maintain constant power requirements can be significantly reduced if a WEC array produces polyphase

power. Additionally, the constant total power is less susceptible to transmission losses for the undersea transmission lines. Finally, polyphase power generation capabilities can significantly reduce the size of the ESS needed for the power management of a MEG.

The work presented here can be expanded to planar irregular waves by designing the WEC array for the significant wave period and then by devising a control strategy to counter the power variation for the waves with smaller significant wave heights. The sinusoidal equations used here to develop the polyphase conditions assumes two dimensional waves. Similar approach based on the summation of sinusoidal series can be used for three-dimensional seas where the argument of the wave elevations has both the $x$ and $y$ coordinates.

**Author Contributions:** Conceptualization, S.H., G.G.P. and W.W.W.; methodology, S.H., G.G.P. and W.W.W.; software, S.H.; validation, S.H. and G.G.P.; formal analysis, S.H., G.G.P. and W.W.W.; writing—original draft preparation, S.H. and G.G.P.; writing—review and editing, S.H., G.G.P. and W.W.W. All authors have read and agreed to the published version of the manuscript.

**Funding:** This research received no external funding.

**Institutional Review Board Statement:** Not applicable.

**Informed Consent Statement:** Not applicable.

**Data Availability Statement:** Not applicable.

**Conflicts of Interest:** The authors declare no conflict of interest.

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
