# Peer review of "Storage Minimization of Marine Energy Grids Using Polyphase Power"

_jmse, doi:10.3390/jmse10020219_

Round 1

Reviewer 1 Report

The topic discussed in this paper is a topical one. Also, is of great interest in electrical power systems field.

In terms of introduction, my suggestion is to approach the topic in more detail. A more detailed bibliographic study is needed , so you can find out (more detailed) what others are doing in this area (topic).  This would result also in a richer and more current (2020, 2021) list of references - the list is short now. Also, my recommendation is to avoid this bibliographic reference style [x-z], when possible. Use it if the papers are on the same topic, and approach these topics with the same methods/models. Otherwise, it is recommended to be discussed separately, possibly making some comparisons with the novelties with which the authors of the current work come.

The research design is presented acceptable, but I would suggest to be slightly improved so as to distinguish very well in each stage what is the paper contribution in the field. In its current form this is very difficult to follow.  For example sections 3 and 4 combined, in the form of Results and Discussions. Thus, it is avoided that in the Results section references are made to certain figures presented in a previous section. In this form the ideas are more difficult to follow and the contribution of the authors as a whole is not seen.

It would be interesting to add in the conclusions section how you see your work further.

I believe that with these changes  the paper may have the expected scientific soundness.

Reviewer 2 Report

This paper presents a study on the use of an array of Wave Energy Converters (WEC) in order to minimize power fluctuations in Marine Energy Grids (MEG). Although the subject is completely outside my area of research, I've felt that the paper is well written and my recommendation will be the approval. Then, I guess the other reviewers can be more critical on the technical aspects of it.

I only feel that, perhaps, if the authors judge it pertinent, a section could be added (before the modelling) with a succinct description of a MEG and WECs. Perhaps some drawings or actual photographs would be great for illustration. 
